# Reagent Effects on the Activated Partial Thromboplastin Time Clot Waveform Analysis: A Multi-Centre Study

**DOI:** 10.3390/diagnostics13142447

**Published:** 2023-07-22

**Authors:** Wan Hui Wong, Chuen Wen Tan, Nabeelah Binti Abdul Khalid, Nadjwa Zamalek Dalimoenthe, Christina Yip, Chaicharoen Tantanate, Rodelio D. Lim, Ji Hyun Kim, Heng Joo Ng

**Affiliations:** 1Department of Haematology, Singapore General Hospital, Singapore 169608, Singapore; tan.chuen.wen@singhealth.com.sg (C.W.T.);; 2Department of Pathology, Hospital Pulau Pinang, Pulau Pinang 10990, Malaysia; 3Department of Haematology, Dr Hasan Sadikin Hospital, Bandung 40161, Indonesia; 4Department of Laboratory Medicine, National University Hospital, Singapore 119074, Singapore; 5Department of Clinical Pathology, Siriraj Hospital, Bangkok 10700, Thailand; 6Institute of Pathology, St Luke’s Medical Centre, Global City 1634, Philippines; 7Department of Pathology, Sengkang General Hospital, Singapore 544886, Singapore

**Keywords:** clot, waveform analysis, coagulation, activated partial thromboplastin time, reagent, reference interval

## Abstract

(1) Background: The activated partial thromboplastin time (APTT)- based clot waveform analysis (CWA) quantitatively extends information obtained from the APTT waveform through its derivatives. However, pre-analytical variables including reagent effects on the CWA parameters are poorly understood and must be standardized as a potential diagnostic assay. (2) Methods: CWA was first analysed with patient samples to understand reagent lot variation in three common APTT reagents: Pathromtin SL, Actin FS, and Actin FSL. A total of 1055 healthy volunteers were also recruited from seven institutions across the Asia-Pacific region and CWA data were collected with the Sysmex CS analysers. (3) Results: CWA parameters varied less than 10% between lots and the linear mixed model analysis showed few site-specific effects within the same reagent group. However, the CWA parameters were significantly different amongst all reagent groups and thus reagent-specific 95% reference intervals could be calculated using the nonparametric method. Post-hoc analysis showed some degree of influence by age and gender with weak correlation to the CWA (r < 0.3). (4) Conclusions: Reagent type significantly affects APTT-based CWA with minimal inter-laboratory variations with the same coagulometer series that allow for data pooling across laboratories with more evidence required for age- and gender-partitioning.

## 1. Introduction

The activated partial thromboplastin time (APTT) is a routine coagulation assay widely performed with automated coagulometers using the optical detection method. Changes to the optical transmittance through platelet-poor plasma (PPP) is detected as the fibrin clot forms upon addition of phospholipids, contact activator, and calcium. The clot time is extracted from the transmittance change, of which the clot kinetics can be represented graphically. APTT-based clot waveform analysis (CWA) is an extended quantitative analysis of this optical transmittance graph that quantifies the waveform characteristics through its derivatives (Figure 1). The first derivative curve, dT/dt, represents the rate of change of transmittance, and the highest value of this curve is the maximum velocity, denoted as min1. The second derivative curve represents the acceleration changes with the highest and lowest points denoted as the maximum acceleration (min2) and deceleration (max2), respectively. As these quantitative parameters interrogate the entire clot formation process, CWA is considered as a global coagulation assay alongside thromboelastography and thrombin generation assay [1].

At present, CWA has demonstrated its potential clinical utility in the assessment of haemophilia and hypercoagulable states [2,3,4,5,6]. A promise of wider adoption by clinicians and laboratories has been enhanced by automated extraction of the data from routine APTT testing. As an APTT-based test, CWA parameters may logically be influenced by the kind of reagents, contact activators, lot changes, and other laboratory factors, as well as the equipment used in its determination. The extent to which these variables influence the reference intervals of CWA parameters within and across laboratories has, however, not been well defined, and is poorly understood. Clarifying these effects will resolve existing uncertainty on its comparability across laboratories and help accelerate CWA standardization as a clinical diagnostic assay [1].

We therefore set about to investigate the reagent effects on the APTT-based CWA obtained from the Sysmex CS series analyzers across different laboratories in the South-east Asia region. We aim to understand the extent of reagent lot change on the CWA. We also hypothesize that reagent types will affect the CWA and that this assay is likely to be robust to inter-laboratory variation with peer groups within 15% of each parameter mean [7]. Our efforts are intended to establish reference intervals that can be easily adopted after local laboratory validation. This paper reports the results of our study.

## 2. Materials and Methods

### 2.1. Inter-Lot Variation

Two different reagent lot numbers were obtained for Pathromtin SL, Actin FS, and Actin FSL. At least 40 patient samples with results across the measuring range were run on the same Sysmex CS2500 (Sysmex Asia-Pacific Pte Ltd, Kobe, Japan) analyzer at the Singapore General Hospital using two different lots of the same reagent in succession. The analyser was rinsed and primed when changing between lots to ensure no carryover. The inter-lot variation of individual CWA parameter is compared by Passing-Bablok regression and expressed as correlation coefficient *r* and percent difference between the lots.

### 2.2. Institutions Involved and Sample Preparation

Seven institutions from across Southeast Asia (Table 1) participated in this study. All institutions are accredited by external accreditation bodies for their prothrombin time (PT) and APTT assays with an average of 3% intra-laboratory variation. Each site collected samples from healthy individuals as part of routine reference interval validation processes. The reference individuals were recruited from clinical and laboratory staff or voluntary donors above age 18 with no prior history of coagulopathy. Subjects who were pregnant or on any anticoagulant or anti-platelet therapy were excluded. Informed consent was obtained from each subject as part of regular validation processes and in accordance with the local institutional ethics regulations before blood sampling was performed.

Blood sampling and processing to obtain the PPP were carried out as per local practices (Table 1) such that the eventual PPP analysed had a platelet count of less than 10 × 10^9^/liter. The PPP samples were all analysed for APTT within 24 h of collection. Any sample with hematocrit exceeding 55%, or obviously hemolysed when inspected visually, was not included in the analysis.

### 2.3. Analyzers Used

CWA data were collected from participating sites using the Sysmex CS series of analyzers. The Sysmex CS 5100 is a high-volume standalone coagulometer while the CS 2100i and CS 2500 analyzers are intermediate-volume desktop analyzers that utilize the same optical detection methods. Individual models differ in terms of throughput and pre-analytical check functions, but the multi-wavelength detection technology and measuring algorithms were the same for the CS series analyzers. Published data have indicated a good correlation between the Sysmex CS 5100 and CS 2000 analyzers with an r = 0.991 [8]. An internal validation between the Sysmex CS 2100i and CS 2500 carried out in Singapore General Hospital have demonstrated that the APTT results were well correlated with an r = 0.991.

### 2.4. Clot Waveform Analysis and Reagent Used

APTT was measured by the Sysmex CS analyzers in the individual sites using specific reagents as outlined in Table 1. All reagents were obtained from Siemens Healthcare (Marburg, Germany) and prepared according to the manufacturer’s recommendations. In this study, contributing sites used one of three types of APTT reagents: Pathromtin SL, Actin FS, and Actin FSL. Pathromtin SL contains silicon dioxide particles (1.2 g/L) as the activator and plant phospholipids (0.25 g/L) [9]. Actin FS contains 1.0 × 10^4^ M ellagic acid and purified soy phosphatides [10]. Actin FSL contains the same concentration of ellagic acid of 1.0 × 10^4^ M and a mixture of soy and rabbit brain phosphatides [11]. Data combined from individual sites were grouped with regard to the reagent used. Quality control was performed at each site during the APTT analysis as per the local guidelines and standard operating procedures. The lot numbers of the reagents were not specified and were dependent on the stock used at each individual site at the time of measurement. The APTT-based CWA data was downloaded for each uncomplicated APTT run. The clot waveform analysis was obtained from a built-in algorithm that gave quantitative CWA parameters in addition to the clot time measured at 660 nm. The CWA parameters obtained were the maximum velocity (min1), maximum acceleration (min2), maximum deceleration (max2) and the corresponding times at which they occurred.

### 2.5. Statistical Analysis

Initial data analysis was done by the Analyse-It for Microsoft Excel software (Version 5.30.1) to check for normality and identify outliers using the Tukey’s method. All outliers were deemed as sporadic analytical errors and were omitted from further analysis. Reagent lot comparability was also performed using Passing-Bablok analysis and Kruskal-Wallis comparison by the Analyse-It software (Version 5.40.2). Subsequent statistical analysis was carried out using SPSS Version 23 (IBM Corporation, Armonk, NY, USA). Site-specific or reagent-specific means were eventually compared taking into account the age and gender effects using a linear mixed model analysis with LSD correction. A *p* < 0.05 was taken as significant. The reagent-specific reference intervals were subsequently calculated using the non-parametric determination recommended by the Clinical & Laboratory Standards Institute (CLSI) guidelines EP28-A3c [12] with the Analyse-It software (Version 5.40.2).

## 3. Results

### 3.1. Reagent Lot Variations

The inter-lot analysis for the three commonly used APTT reagents (Pathromtin SL, Actin FS, and Actin FSL) showed good correlation of all CWA parameters between two reagent lots across the normal and pathological result ranges. The correlation coefficients for the CWA parameters were all greater than 0.95 and the percent inter-lot differences were within 10% (Table 2), suggesting result consistency despite lot changes. There was no statistical significance difference between the median results for all CWA parameters for the two different lots of all three reagents.

### 3.2. Distribution of the Reference Population

A total of 1055 healthy individuals were recruited at the seven participating sites. Although some sites did not submit age and gender details, the CWA data were used in the analysis without exclusion. The data collected were first analysed for outliers for each individual parameter. Outliers identified were investigated for sample collection and processing errors at each site, but none were found. The number of outliers eliminated from the final analysis for each parameter at each site was not more than 10% of the numbers collected, and they were subsequently removed for the final analysis. The total numbers of data points included for the APTT-based CWA reference distribution parameters at each site are shown in Table 3.

Age and gender details were available from 757 subjects ranging from 18 to 72 years. There were 273 males (mean age 32.13 years, standard deviation (SD) 10.20 years) and 484 females (mean age 33.70 years, SD 10.28 years). There were disproportionately more females than males, but the mean ages were minimally different (*p* = 0.043).

The data collected were also analysed for normality by the Shapiro-Wilk test and visually by the histograms (Figure 2). Although non-normality was observed in some data from individual sites, the histograms did not show remarkable skewness compared to the normal distribution lines. The means and medians calculated for each site and reagent type are reported in Table 3.

### 3.3. Reagents and Sites Differences and the Corresponding CWA Reference Intervals

The site-specific variation was investigated using the linear mixed model analysis to establish if data could be combined from different sites. The effects of site on the individual CWA parameter, including the clot time, were not significant (all *p* > 0.05), suggesting little site-specific variation within the same reagent group (Figure 3a). As such, data from different sites were combined for subsequent analysis.

The means of the CWA data collected from each site and as a combined reagent group are shown in Table 3. Since the secondary analysis showed some age and gender influence, the adjusted means of the different reagent groups are shown in Figure 3b and compared pairwise with the LSD correction.

The clot times differed significantly between the reagent groups, verifying that the APTT was highly dependent on the reagent type. The CWA parameters were also significantly different between all the reagent groups, with the exception of Max2, in which the Actin FS and Actin FSL did not differ.

The corresponding CWA 95% reference intervals evaluated by the nonparametric method were also calculated as reflected in the 2.5th (lower limit) and 97.5th (upper limit) percentile values (Table 3). Since the nonparametric method recommended by the CLSI guidelines was not affected by the statistical distribution or physiological variation, no transformation was performed despite slight non-normality. The 90% confidence intervals for the limits for individual sites with sample sizes of less than 120 were calculated by the bootstrap method. Although sites which recruited less than 120 subjects were expected to have wider intervals, these calculated intervals did not differ very much from the interval calculated for the combined reagent group. This affirmed that there was little site-specific variation within the same reagent type. The reference intervals for CWA parameters were drastically lower for the Pathromtin SL group compared to both the Actin FS and Actin FSL groups.

### 3.4. Influence of Age and Gender

In post-hoc analysis to elucidate how the CWA parameters were affected by different reagents, age and gender were used as independent variables in a mixed model analysis of the parameter (dependent variable). Both variables significantly affected all the CWA parameters but did not show any site-specific differences (Appendix A).

The clot times for each reagent were not significantly influenced by age or gender. In comparison, the CWA parameters and the corresponding times at which they occurred were all significantly influenced by age (Appendix A). The CWA parameters (Min1, Min2, and Max2) were statistically different between genders but not significantly different for the times at which the CWA parameters (TMin1, TMin2, and TMax2) occurred. Despite the significant influence of age in the mixed model, the correlations between age and the CWA parameters are weak (r < 0.3).

The possibility of separating reference intervals for the age and gender subclasses were considered given that these factors significantly affected the CWA parameters. The CLSI guidelines suggested several methods to partition reference intervals for various subclasses, but partitioning should not be estimated unless the differences between the subclass means are at least 25% as large as the 95% reference interval estimated from the combined sample of reference subjects. Each subclass should have at least 120 subjects for such comparisons to be valid. Although we did not recruit 120 subjects in each 10-year subclass, the observed means of the age and gender subclasses were nonetheless compared. The differences in means were within 25% of the 95% reference interval, except for some subclasses aged greater than 50. In such subclasses, the numbers of subjects were very small and might not truly estimate the observed means.

## 4. Discussion

In this multi-centre regional study conducted on the Sysmex CS platform, the three commonly-used reagents differed in their activator type (silica—Pathromtin SL, ellagic acid—Actin FS and Actin FSL) and phospholipid content, with Actin FS having lower phospholipids compared to Actin FSL. Our results showed that the APTT-based CWA, like the APTT, is sensitive to the activator rather than the phospholipid content of the reagent. This is evident as CWA parameters obtained from the Pathromtin SL varied quite significantly compared to those obtained from Actin FS and Actin FSL, while little variability was observed between CWA results for Actin FS and Actin FSL.

Despite the inevitable differences in local laboratory practices, our data showed that there was very little inter-laboratory variability for results obtained within each reagent type and lot change. The current recommended laboratory processes for obtaining the CWA appears robust in ensuring consistency of results and was only susceptible to the type of reagent employed. This has enabled the pooling of data to calculate universal reagent-specific reference intervals that could be shared amongst users of the same reagent-analyzer combinations after local validation procedures.

The reference intervals (RIs) of the CWA parameters for the study population were calculated from a wide spread of individuals across several countries, and we believe these can be applied to the general adult population regardless of age or gender. Although noticeable differences in the CWA parameters exist between specific age subclasses such as the youngest and oldest strata, the outlier numbers in these subclasses were only a small fraction of the total recruited population. This negates the value of partitioned RIs according to either age or gender^12^. Furthermore, the small number of recruited subjects in these extreme age groups would have limited the accuracy of individual subclass RIs. Currently, the clinical significance of age and gender differences on the RI is uncertain. It is plausible that proven age-dependent fluctuations in coagulation factors and other contributing physiological changes could lead to parallel fluctuations in the CWA. Until more data emerge on the clinical impact of age or gender stratification, we are of the view that a common RI for the CWA parameters best serves its purpose in clinical and research practices.

Our study has benefitted from the collaborative efforts of participating laboratories in different countries, which has allowed us to investigate the robustness of the platform against varied operating environments. Consistent results for the same reagent–analyzer combination means that CWA data collection can be expanded across different sites and combined to accelerate collaborative efforts in clinical and research protocols. Our results, however, cannot be generalized to CWA obtained with other optical detection analyzers. We also did not study the effect of different reagent combinations on PT-based CWA, which has less defined clinical utility. Our assessment of the effects of age and gender would also have benefitted from a larger study population with a better spread of subjects across different age sub-classes.

## 5. Conclusions

We have demonstrated that the type of reagent used significantly affects APTT-based CWA, but there are minimal inter-laboratory and inter-lot variations between laboratories employing the same reagent-analyzer set-up. A common set of reference intervals established for individual reagents can hence be used for cross-referencing in quality assurance programs and provide confidence for combining data in multi-centre clinical and research protocols to further simplify and spur greater utilization of CWA in daily practice.

## Figures and Tables

**Figure 1 diagnostics-13-02447-f001:**
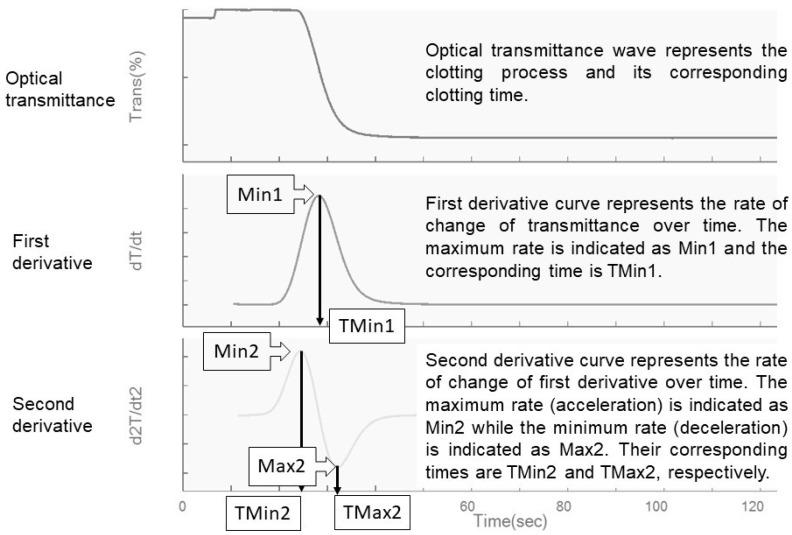
APTT-based clot waveform analysis (CWA) and its derivatives.

**Figure 2 diagnostics-13-02447-f002:**
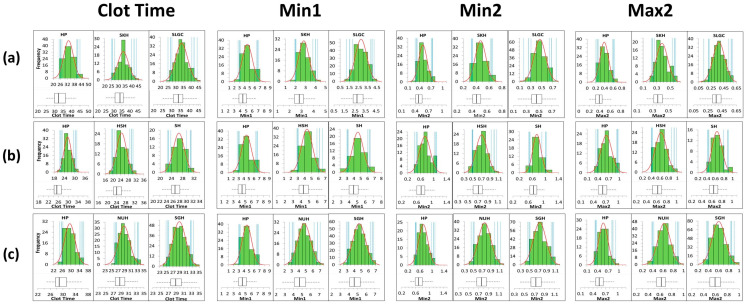
Histograms of all measured parameters compared to the normal distribution lines as well as the reference intervals with the 90% confidence intervals. Institutions using (**a**) Pathromtin SL, (**b**) Actin FS, and (**c**) Actin FSL.

**Figure 3 diagnostics-13-02447-f003:**
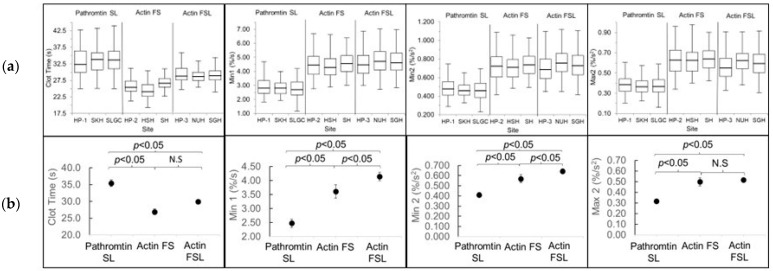
(**a**) Boxplots of CWA parameters showed no significance based on the site. (**b**) Comparison of the adjusted (adjusted for age and gender) means (black circle) with the standard errors between the different reagent groups for Min1, Min2, and Max2. A *p*-value of <0.05 is taken as significant, otherwise not significant (N.S).

**Table 1 diagnostics-13-02447-t001:** List of institutions involved with the analytical parameters and conditions.

	HP	HSH	NUH	SH	SGH	SLGC	SKH
Institution name, country	Hospital Pulau Pinang, Malaysia	Hasan Sadikin Hospital, Indonesia	National University Hospital, Singapore	Siriraj Hospital, Thailand	Singapore General Hospital, Singapore	St. Luke’s General Hospital, The Philippines	Sengkang General Hospital, Singapore
Analyzer	CS2500	CS2100i	CS5100	CS2100i	CS2100i	CS2100i	CS2500
APTT reagent	Pathromtin SL (*n* = 94) Actin FS (*n* = 90) Actin FSL (*n* = 93)	Actin FS (*n* = 79)	Actin FSL (*n* = 140)	Actin FS (*n* = 46)	Actin FSL (*n* = 257)	Pathromtin SL (*n* = 174)	Pathromtin SL (*n* = 82)
Average % CV for APTT assay	Pathromtin SL: 1.6% Actin FS: 0.9% Actin FSL: 1.0%	3.8%	1.5%	0.9%	0.6%	5.4%	1.2%
Blood drawn through	Vacutainer	Vacutainer	Vacutainer	Vacutainer	Vacutainer	Vacutainer	Vacutainer
Blood tubes (manufacturer)	3.2% sodium citrate (Becton-Dickinson Company, Franklin Lakes, NJ, USA)						
Sample processing	4000 rpm, 10 min	4000 rpm, 10 min	2800× *g*, 10 min	1500× *g*, 15 min	3000× *g*, 15 min	1500× *g*, 10 min	6000 rpm, 3 min
External quality assurance accreditation	RCPA ^2^	UK IEQAS ^3^	CAP ^1^	CAP	CAP	CAP	CAP participation only

^1^ College of American Pathologists. ^2^ Royal College of Pathologists Australasia. ^3^ United Kingdom International External Quality Assurance Scheme.

**Table 2 diagnostics-13-02447-t002:** Inter-lot reagent differences obtained from the same sample by one analyser, expressed as range of sample results, correlation coefficient *r,* and percent difference (% difference) between the lots.

Reagent	Lot	Clot Time, s	Min1, %/s	TMin1, s	Min2, %/s^2^	TMin2, s	Max2, %/s^2^	TMax2, s
Pathromtin SL (*n* = 47)	1	24.60–58.10	1.292–6.721	24.50–56.20	0.214–1.165	20.80–51.50	0.111–1.005	28.00–63.50
	2	24.90–57.30	1.280–6.710	24.90–56.70	0.220–1.176	21.30–52.00	0.116–1.024	28.30–62.80
	r	0.982	0.998	0.98	0.997	0.98	0.997	0.98
	*p*-value	0.291 (N.S)	0.472 (N.S)	0.261 (N.S)	0.390 (N.S)	0.260 (N.S)	0.518 (N.S)	0.431 (N.S)
	% difference	1.10%	0.20%	1.50%	0.90%	1.90%	4.20%	0.90%
Actin FS (*n* = 57)	1	23.10–50.30	2.814–8.192	23.00–49.00	0.421–1.378	19.30–44.30	0.300–1.151	26.70–53.80
	2	23.40–53.10	2.809–8.245	23.30–51.90	0.427–1.362	19.60–47.20	0.301–1.156	26.80–56.60
	r	0.978	0.998	0.976	0.995	0.976	0.994	0.977
	*p*-value	0.147 (N.S)	0.475 (N.S)	0.173 (N.S)	0.364 (N.S)	0.288 (N.S)	0.475 (N.S)	0.274 (N.S)
	% difference	0.30%	0.70%	0.30%	1.90%	0.10%	2.50%	0.50%
Actin FSL (*n* = 44)	1	19.10–69.00	0.949–8.419	19.20–68.40	0.118–1.375	15.70–63.10	0.087–1.205	22.50–74.00
	2	19.10–63.90	1.036–8.543	19.20–63.30	0.136–1.399	15.80–58.20	0.099–1.230	22.60–68.50
	r	0.997	0.998	0.997	0.995	0.997	0.995	0.997
	*p*-value	0.306 (N.S)	0.471 (N.S)	0.268 (N.S)	0.390 (N.S)	0.305 (N.S)	0.429 (N.S)	0.306 (N.S)
	% difference	3.60%	4.60%	3.40%	7.60%	3.40%	8.80%	3.50%

**Table 3 diagnostics-13-02447-t003:** Unadjusted and adjusted (adjusted for age and gender) means with the percentile values of the CWA parameters of the reference population. A *p*-value of <0.05 is taken as significant, otherwise not significant (N.S).

Reagent	Pathromtin SL				Actin FS				Actin FSL			
Site	HP	SKH	SLGC	Total	HP	HSH	SH	Total	HP	NUH	SGH	Total
Clot Time, s												
Number used	94	80	171	345	89	77	45	211	93	140	255	482
Unadjusted mean	33.00	33.59	33.82	33.54	25.82	24.45	26.65	25.49	29.49	28.88	28.95	28.96
2.5th percentile (90% CI)	25.98 (25.00–27.20)	25.51 (25.10–27.41)	26.63 (25.00–27.50)	26.30 (25.20–26.90)	21.91 (21.30–22.68)	19.76 (19.30–20.99)	23.00 (22.70–23.76)	21.09 (19.30–21.50)	25.67 (24.80–26.27)	26.35 (25.40–26.50)	25.44 (24.90–25.70)	25.60 (25.40–26.00)
97.5th percentile (90% CI)	40.97 (39.96–42.80)	41.76 (39.60–43.30)	41.95 (40.50–44.00)	41.15 (40.50–43.30)	30.26 (29.45–31.20)	29.72 (28.44–30.40)	30.67 (29.36–31.00)	30.24 (29.40–31.00)	31.19 (34.15–35.90)	33.25 (32.20–33.40)	32.50 (32.40–33.30)	33.19 (32.50–33.30)
Adjusted mean ± SE				35.40 ± 0.99				26.92 ± 0.84				29.92 ± 0.33
Site-specific comparisons	*p* = 0.211 (N.S)											
**Min1,** **%** **/s**												
Number used	91	78	171	343	88	79	44	211	91	140	255	486
Unadjusted mean	2.914	2.809	2.775	2.833	4.534	4.341	4.574	4.470	4.571	4.773	4.684	4.704
2.5th percentile (90% CI)	1.931 (1.820–2.078)	2.002 (1.933–2.044)	1.533 (1.172–1.857)	1.792 (1.516–1.919)	2.981 (2.749–3.186)	2.991 (2.890–3.103)	3.046 (2.991–3.211)	2.993 (2.890–3.092)	3.069 (2.982–3.222)	3.086 (2.717–3.306)	3.129 (2.908–3.252)	3.118 (3.031–3.202)
97.5th percentile (90% CI)	4.407 (4.209–4.697)	3.884 (3.637–3.998)	3.977 (3.814–4.197)	4.012 (3.940–4.278)	6.571 (6.335–6.675)	6.176 (5.787–6.616)	6.236 (5.986–6.374)	6.414 (6.227–6.626)	6.697 (6.399–6.857)	6.640 (6.166–7.022)	6.521 (6.261–6.700)	6.555 (6.373–6.796)
Adjusted mean ± SE				2.477 ± 0.145				3.609 ± 0.241				4.144 ± 0.150
Site-specific comparisons	*p* = 0.113 (N.S)											
**Time of min1 (Tmin1), s**												
Number used	94	80	168	341	89	77	45	211	93	140	255	483
Unadjusted mean	32.93	33.45	33.54	33.36	25.68	24.40	26.51	25.39	29.15	28.64	28.69	28.70
2.5th percentile (90% CI)	26.03 (25.20–27.28)	25.55 (25.20–27.32)	26.70 (25.10–27.50)	26.26 (25.20–27.00)	21.91 (21.30–22.70)	19.80 (19.30–20.99)	22.90 (22.60–23.75)	21.09 (19.30–21.60)	25.60 (24.70–26.20)	26.25 (25.40–26.40)	25.34 (24.80–25.60)	25.51 (25.30–25.90)
97.5th percentile (90% CI)	40.52 (39.20–42.20)	41.22 (39.30–42.80)	40.58 (40.00–41.60)	40.55 (40.10–41.30)	29.95 (29.15–30.90)	29.49 (28.38–30.20)	30.35 (28.90–30.70)	29.94 (29.20–30.70)	34.57 (33.20–35.30)	32.80 (31.80–33.00)	32.14 (31.80–32.80)	32.67 (31.90–32.80)
Adjusted mean ± SE				35.88 ± 1.02				27.90 ± 0.88				29.59 ± 0.32
Site-specific comparisons	*p* = 0.131 (N.S)											
**Min2,** **%/s^2^**												
Number used	91	78	172	339	88	78	44	210	91	140	255	486
Unadjusted mean	0.488	0.467	0.464	0.471	0.732	0.707	0.737	0.724	0.710	0.758	0.740	0.740
2.5th percentile (90% CI)	0.319 (0.289–0.343)	0.333 (0.328–0.342)	0.268 (0.230–0.305)	0.276 (0.261–0.291)	0.463 (0.414–0.404)	0.490 (0.484–0.504)	0.505 (0.493–0.532)	0.440 (0.412–0.468)	0.460 (0.448–0.478)	0.496 (0.451–0.520)	0.492 (0.464–0.518)	0.484 (0.465–0.506)
97.5th percentile (90% CI)	0.727 (0.700–0.744)	0.643 (0.619–0.653)	0.656 (0.640–0.697)	0.667 (0.652–0.682)	1.069 (1.029–1.089)	0.990 (0.9230–1.057)	1.010 (0.960–1.028)	1.008 (0.980–1.036)	1.056 (0.99–1.098)	1.060 (0.985–1.119)	1.041 (1.007–1.062)	1.042 (1.021–1.075)
Adjusted mean ± SE				35.88 ± 1.02				27.90 ± 0.88				29.59 ± 0.32
Site-specific comparisons	*p* = 0.113 (N.S)											
**Time of min2 (Tmin2), s**												
Number used	94	79	168	341	89	77	45	211	91	140	254	483
Unadjusted mean	29.19	29.55	29.79	29.57	21.87	20.68	22.68	21.61	24.98	24.74	24.78	24.78
2.5th percentile (90% CI)	22.46 (21.70–23.79)	22.00 (21.60–23.50)	23.20 (21.60–23.90)	22.71 (21.70–23.50)	18.30 (17.80–18.90)	16.37 (15.90–17.30)	19.26 (18.90–20.03)	17.36 (15.90–18.00)	21.78 (21.00–22.45)	22.45 (21.90–22.80)	21.64 (21.50–22.00)	21.81 (21.60–22.20)
97.5th percentile (90% CI)	36.42 (35.61–38.00)	36.33 (35.30–37.20)	36.68 (35.90–37.50)	36.45 (36.00–37.10)	25.74 (25.00–26.60)	25.48 (24.81–26.20)	26.24 (25.00–26.50)	25.70 (25.00–26.50)	28.63 (28.10–29.20)	28.40 (27.60–28.80)	27.90 (27.50–28.30)	28.20 (27.90–28.60)
Adjusted mean ± SE				32.07 ± 0.98				23.97 ± 0.82				25.61 ± 0.29
Site-specific comparisons	*p* = 0.126 (N.S)											
**Max2,** **%/s^2^**												
Number used	92	81	172	342	88	79	44	211	90	140	255	485
Unadjusted mean	0.394	0.377	0.372	0.378	0.634	0.624	0.638	0.631	0.561	0.619	0.603	0.600
2.5th percentile (90% CI)	0.230 (0.204–0.255)	0.237 (0.225–0.264)	0.212 (0.159–0.220)	0.218 (0.208–0.234)	0.379 (0.340–0.408)	0.416 (0.401–0.429)	0.433 (0.421–0.453)	0.402 (0.378–0.421)	0.350 (0.327–0.371)	0.397 (0.382–0.418)	0.385 (0.365–0.404)	0.376 (0.361–0.387)
97.5th percentile (90% CI)	0.592 (0.568–0.605)	0.560 (0.532–0.574)	0.567 (0.528–0.587)	0.571 (0.559–0.587)	0.949 (0.923–0.962)	0.930 (0.866–0.977)	0.887 (0.793–0.903)	0.944 (0.902–0.962)	0.834 (0.800–0.908)	0.878 (0.847–0.906)	0.857 (0.833–0.905)	0.858 (0.838–0.891)
Adjusted mean ± SE				0.316 ± 0.021				0.498 ± 0.038				0.515 ± 0.021
Site-specific comparisons	*p* = 0.05 (N.S)											
**Time of max2 (Tmax2), s**												
Number used	94	80	168	342	89	77	45	211	93	140	253	481
Unadjusted mean	36.66	37.22	37.27	37.09	29.51	28.11	30.34	29.18	33.22	32.53	32.57	32.62
2.5th percentile (90% CI)	29.58 (28.50–30.89)	29.03 (28.60–31.01)	30.20 (28.40–31.10)	29.80 (28.70–30.50)	25.24 (24.70–26.30)	23.29 (22.70–24.40)	26.54 (26.20–27.40)	24.49 (22.80–25.20)	29.19 (28.40–29.77)	29.90 (28.90–30.10)	29.00 (28.40–29.30)	29.11 (28.90–29.60)
97.5th percentile (90% CI)	44.80 (43.60–46.01)	45.57 (43.20–47.10)	44.50 (44.10–46.00)	44.64 (44.20–45.60)	34.31 (33.43–35.30)	33.58 (32.53–34.10)	34.61 (33.16–35.00)	34.10 (33.30–35.00)	37.39 (36.81–37.90)	37.20 (36.00–37.50)	36.37 (36.10–37.10)	36.90 (36.30–37.30)
Adjusted mean ± SE				39.92 ± 1.09				25.93 ± 1.40				33.66 ± 0.72
Site-specific comparisons	*p* = 0.121 (N.S)											

## Data Availability

No new data were created or analyzed in this study. Data sharing is no applicable to this article.

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
