# Peer review of "Reagent Effects on the Activated Partial Thromboplastin Time Clot Waveform Analysis: A Multi-Centre Study"

_diagnostics, 2023, doi:10.3390/diagnostics13142447_

Round 1

Reviewer 1 Report

The topic is interesting and well presented. You just need to add p value to the tables.

Author Response

Thank you for the comments. We have made some edits based on your comments:

1. Line 142: p-value for correlation coefficients

We have added the p-values for the inter-lot comparisons with an additional line "There was no statistical significance difference between the median results for all CWA parameters for the two different lots of all three reagents." and included the respective p-values in Table 2. 

2. Line 170: p-values for site-specific comparisons

We have added the p-values in Table 3.

For your consideration please.

Reviewer 2 Report

The paper by Wan Hui Wong and coworkers showed that reagent type significantly affects APTT-based CWA with minimal inter-laboratory variations with the same coagulometer series that allow for data pooling across laboratories with more evidence required for age- and gender-partitioning. Authors came to their conclusions by performing their study on analyzers across different laboratories in the South-east Asia region which increases the possibility of clinical use of the presented results. The paper is well organized and the results add important information to the field. However I have to point out on some editorial errors that have occurred in this work. 

Minor comments:

1. The current structure of the publication is asymmetrical and therefore unacceptable - there is too much margin on the left side of each page.

2. There are to many text breaks (spaces) in the sections: Introduction, Results and Discussion. 

3. The results are not presented in a clear way - I suggest presenting the results in the form of graphs instead of tables.

Author Response

Thank you for the comments. We have made some edits based on your comments:

1.The current structure of the publication is asymmetrical and therefore unacceptable - there is too much margin on the left side of each page.

The publication was structured based on the format provided by the journal hence we are not able to change the page margins. 

2. There are to many text breaks (spaces) in the sections: Introduction, Results and Discussion. 

The spaces between the sections had been reduced. 

3. The results are not presented in a clear way - I suggest presenting the results in the form of graphs instead of tables.

We have included a Figure 3 showing the boxplots of the CWA parameters by site and adjusted means of the reagents. Table 2 was left in the paper to provide the reference intervals for which individual sites might find useful. 

For your consideration please.

Reviewer 3 Report

This is an exciting research paper.

However, a few suggestions are placed to further improve the manuscript.

Introduction:

Comment 1: Nicely written. However, it will be nice to incorporate a Figure and a reference for all the discussion at the end of the first paragraph.

Method:

Comment 2: Was IEC nod taken (if yes, please mention the no). Was this study registered?

Results:

Comment 3: Looks good

Discussion:

Comment 4: First paragraph may be deleted or shifted to the introduction.

Comment 5: Conclusion should be more precise

Reference:

Comment 6: adequate

Table and Figure:

Comment 7: looks good

Author Response

Thank you for the comments. We have made some edits based on your comments:

Comment 1: Nicely written. However, it will be nice to incorporate a Figure and a reference for all the discussion at the end of the first paragraph.

We have added a Figure 1 to show the clot waveform analysis and included a reference at the end of the first paragraph.

Comment 2: Was IEC nod taken (if yes, please mention the no). Was this study registered?

The IEC was taken by individual sites based on their local institutional ethics regulations for pre-service evaluation of reference intervals. We have edited Line 89 for clarity.

Comment 3: First paragraph of discussion may be deleted or shifted to the introduction.

First paragraph of discussion had been deleted.

Comment 4: Conclusion should be more precise

We have edited the conclusion for better clarity.

For your consideration please.

Round 2

Reviewer 2 Report

Authors responded to all my comments. I do not see further objections and I accept revised version of the manuscript on the present form.